# Gastrointestinal Involvement in Dermatomyositis

**DOI:** 10.3390/diagnostics12051200

**Published:** 2022-05-11

**Authors:** Ana Matas-Garcia, José C. Milisenda, Gerard Espinosa, Míriam Cuatrecasas, Albert Selva-O’Callaghan, Josep María Grau, Sergio Prieto-González

**Affiliations:** 1Muscle Research Unit, Department of Internal Medicine, Hospital Clínic de Barcelona, Universidad de Barcelona, Center for Biomedical Research on Rare Diseases (CIBERER), 08036 Barcelona, Spain; anmatas@clinic.cat (A.M.-G.); jcmilise@clinic.cat (J.C.M.); jmgrau@clinic.cat (J.M.G.); 2Department of Autoimmune Diseases, Hospital Clínic de Barcelona, Universitat de Barcelona, 08036 Barcelona, Spain; gespino@clinic.cat; 3Department of Pathology, Hospital Clínic de Barcelona, 08036 Barcelona, Spain; mcuatrec@clinic.cat; 4Systemic Autoimmune Diseases Unit, Hospital Universitari Vall d’Hebron (HVH), Universitat Autònoma de Barcelona, 08035 Barcelona, Spain; aselva@vhebron.net

**Keywords:** dermatomyositis, gastrointestinal involvement, gastrointestinal perforation, gastrointestinal bleeding, intestinal pneumatosis

## Abstract

Dermatomyositis is a systemic vasculopathy mainly affecting skin, muscle and lung, but may affect the gastrointestinal tract. We aim to describe clinical characteristics of patients with severe gastrointestinal involvement related to dermatomyositis in our center and medical literature. We retrospectively analysed these patients in our center, including cases of erosions/ulcers, perforation or digestive bleeding. Reported cases from April 1990 to April 2021 were reviewed through PubMed and Cochrane. From our cohort (*n* = 188), only 3 presented gastrointestinal compromise. All were women (10, 46 and 68 years). The initial symptom was abdominal pain and all had ≥2 episodes of digestive bleeding. All died due to complications of gastrointestinal involvement. Available pathological samples showed vascular ectasia. From the literature review (*n* = 50), 77% were women with a mean age of 49 years and the main symptom was abdominal pain (65%). All presented active muscular and cutaneous involvement at complication diagnosis. Mortality was 41.7%. The underlying lesion was perforation or ulcer (*n* = 22), intestinal wall thickening (*n* = 2), macroscopic inflammation (*n* = 2) or intestinal pneumatosis (*n* = 15). In 13 cases, vasculitis was described. Gastrointestinal involvement in dermatomyositis denotes severity, so an early intensive treatment is recommended. Pathological findings suggest that the underlying pathophysiological mechanism is a vasculopathy and not a true vasculitis.

## 1. Introduction

Dermatomyositis (DM) is a subtype of idiopathic inflammatory myopathy characterised by the presence of inflammatory infiltrates in proximal skeletal muscles and the skin [1]. Although certain pathophysiology of DM is unknown, several genetic, immunologic, and environmental factors are implicated in this disease. Roughly, it is thought to be the result of a humoral-mediated attack directed against the muscle capillaries and the endothelium of arterioles. This condition is seen in both children and adults, with a female to male predominance of about two to one [2]. Extramuscular manifestations may be present to a varying degree, including lung, articular, heart, genitourinary, nervous system and gastrointestinal (GI) tract involvement [3].

It is known that GI tract involvement is rare in patients with DM, with only a few reported cases in medical literature, but it may be life-threatening. As a general rule, GI involvement in cases of adult DM is usually mild and occurs as a result from damage to the smooth muscle of the myoenteric system, leading to motility disorders. GI tract involvement usually occur late in the course of the disease, and not always during highly disease activity. Late onset of abdominal symptoms may be masked by the ongoing immunosuppressive treatment, and long-term treatment with high doses of corticosteroids may also be a predisposing factor to the develop GI complications [4].

The most common GI manifestation in DM is dysphagia (about 2/3 of DM patients with active disease) as a result of pharyngeal and upper oesophageal compromise. However, it can be argued whether dysphagia should be regarded as a muscle manifestation or an extramuscular manifestation. Patients may occasionally present with ulceration, haemorrhage or perforation of the GI tract, as well as intestinal pneumatosis (IP) [5]. For this reason, progressive, persistent or recurrent abdominal pain should be also considered a significant symptom, especially in juvenile DM [4].

An early diagnosis of intestinal ischaemia or necrosis may be challenging [1,6]. There are no laboratory test or clinical markers available to properly assess disease activity. Indeed, serum levels of CRP or muscle enzymes are not directly related to intestinal involvement [7]. Radiographic findings on plain and contrast abdominal studies are usually normal or nonspecific, even in the presence of mild bowel perforation, with spontaneous closure or contained by adjacent structures (extraluminal air is visible only in 50–70% of patients [3]).

The aim of this study was to describe clinical and histologic characteristics of patients with severe GI involvement related to DM in our center and medical literature.

## 2. Materials and Methods

Here, we report three cases of DM with GI involvement from our cohort of DM patients attended at the Hospital Clínic de Barcelona (HCB), a referral center for muscle diseases, from April 1990 to April 2021. We also analysed the reported cases of severe GI involvement using PubMed/Medline and Cochrane (both in English and Spanish language) in the same period of time. Data was collected using standardised prospective data collection. A flow diagram of the included studies is presented in the (Appendix A). Diagnosis of DM was established following the American College of Rheumatology/the European League Against Rheumatism (ACR/EULAR) 2013 classification criteria. Additionally, severe GI involvement was defined as the presence of one of the following in the absence of another more plausible etiology: erosions or ulcers, perforation (duodenal, gastric, or intestinal), IP and upper or lower digestive haemorrhage (DH) or in which a fatal outcome occurred including death or ICU admission. The reason for focusing attention on severe GI manifestations and not in all GI involvement was to respond especially to those scenarios that are often life-threatening, for which there is a lack of knowledge. In addition, disease activity was assessed using the Disease Activity Score (DAS) and IMACS criteria.

This study was approved by the Hospital Clínic de Barcelona Institutional Review Board, and written informed consent was obtained from each participant. Every muscle biopsy was performed exclusively for diagnostic purposes by two myology experts (JMG and JCM) at HCB.

## 3. Results

Three patients with DM and severe GI manifestations were identified out of 188 patients reviewed from our cohort. Following is a case-by-case description of this patients. The results of the three patients are summarised in Table 1 at the end of the section (Table 1).

### 3.1. Case 1

A 10-year-old female patient with no medical history, presented with 1-year history of erythematous lesions in the malar region, Gottron’s papules and proximal muscle weakness, inability to walk and dysphagia to solids. Electromyography revealed a myositic pattern and a muscle biopsy showed perivascular infiltrates and perifascicular atrophy. In semithin sections, a loosening of peripheral capillaries together with dilatation of central capillaries was clearly seen (Figure 1A). Prednisone (PDN) and cyclosporine (CyA) were started, achieving clinical remission.

Eleven months after the diagnosis of DM, on treatment with PDN 5 mg/day and CyA 1.5 mg/kg/day, she presented to the emergency department with refractory epigastric pain and oral intolerance, as well as worsening of the skin lesions (Figure 2). A fibrogastroscopy (FGS) evidenced an inflammatory-appearing mucosa with multiple superficial erosions. Work-up for Helicobacter pylori and other bacterial and viral pathogens was negative. Treatment with IGIV was initiated, with clinical improvement and resolution of the erosions was observed in sequential FGS. These lesions were attributed to myositis due to the existence of disease activity and after excluding other unrelated pathologies.

After her discharge, the patient continued to complain episodic epigastralgia and mild weakness, which were believed to reflect exacerbation of the GI inflammation. Thus, ten months later, she was again admitted due to abdominal pain and rectal bleeding. Due to haemodynamic instability, an emergency surgery was performed, showing a perforated duodenal ulcer at the duodenal second portion. Despite the surgical treatment, she finally died at the pediatric ICU. The family declined a clinical autopsy.

### 3.2. Case 2

A 68-year-old woman with a prior medical history of dyslipidaemia on statin therapy, hypertension and colon adenocarcinoma, treated with surgery and chemotherapy three years after. She presented with a three-week history of proximal weakness in shoulder and pelvic girdle and inability to walk. A muscle biopsy revealed perivascular inflammatory infiltrates and perifascicular atrophy so diagnosis of DM was established. She was treated with PDN and achieved clinical remission. Six months later, she was admitted with diffuse abdominal pain, watery diarrhoea and oral intolerance. A fibrocolonoscopy (FCS) evidenced multiple superficial ulcers covered with fibrin. Mesalazine was then added to treatment.

Two months later, she was admitted for severe abdominal pain and haematemesis. FGS evidenced superficial ulcers covered with fibrin from middle third of the oesophagus to the second portion of the duodenum). Initial endoscopic improvement was elucidated after initiation of immunosuppression. No other risk factors for peptic ulcer disease, such as NSAIDs or smoking were present. Cultures and immunostaining for bacterial and viral pathogens, including CMV virus, herpes simplex virus, enteric bacteria, Clostridium difficile toxin assay and rapid urease test for H. pylori were all negative. However, a chest CT scan showed extensive empyema and mediastinitis, so she underwent an emergency surgery. Oesophagectomy was performed due to an oesophageal perforation and the histopathological study revealed marked atrophy, oedema and slight fibrosis of the internal muscle layer (Figure 1B,C).

Six months after, she was readmitted with rectal bleeding. Treatment with IGIV (5 doses of 400 mg/kg/day) and methylprednisolone (3 doses of 1 g/day) was started, but she finally died three weeks later.

### 3.3. Case 3

A 46-year-old woman was diagnosed with DM at age of 15 for a severe axial and proximal muscle weakness, heliotrope rash and erythematous lesions in photoexposed areas. Two decades after the diagnosis, being in clinical remission on treatment with 5 mg of prednisone every other day, she presented a spontaneous jejunal perforation, treated by simple closure. Five years later, she presented a muscle flare associated with diffuse abdominal pain and rectal bleeding, so she was admitted to the ICU. An FGS showed multiple ulcers on second and third duodenal portion. An abdominal CT scan revealed a duodenal perforation, so she was treated with methylprednisolone (1 g/day bolus dose, 3 doses), plasma apheresis and IGIV (400 mg/kg bolus dose, 5 doses). Once clinical stabilisation was achieved, she was kept in maintenance treatment with azathioprine and prednisone 5 mg/day for 5 years.

After this period, she was readmitted for epigastralgia, oral intolerance and abundant rectal bleeding. An FGS evidenced an erythematosus mucosa in the antrum and in the body of the stomach (Figure 1D), and multiple ulcers covered with fibrin in the duodenum, which extended from the bulb to the third portion. Work-up for Helicobacter pylori and other bacterial and viral pathogens was negative. Non-specific duodenitis with fibrosis of lamina propia, but without atrophy, and marked vascular ectasia were described in the pathological study. Although vascular ectasia is more common in patients with scleroderma it was not the scenario of this patient. Treatment with IGIV (5 consecutive days of 400 mg/kg) and methylprednisolone (3 doses of 1 g/day) was started, and the patient was admitted to ICU.

An arteriography demonstrated active bleeding in the jejunal branch of the superior mesenteric artery, which was selectively embolised, followed by clinical improvement and discharged to continue outpatient treatment with IGIV cycles every 4 weeks. After 10 months, IGIV was switched to azathioprine. A few months later, tacrolimus and rituximab were initiated because of a refractory disease, achieving a clinical remission. Three years later, the patient died at home due to a sudden cardiorespiratory arrest. A judicial autopsy was carried out, but the report was not available. Due to the atypical GI involvement, especially the arterial bleeding, Behçet’s disease was considered and ruled out.

## 4. Discussion

In the past 30 years, a total of 188 patients have been diagnosed with DM in our unit, and 3 of them developed severe GI involvement. Moreover, we have identified an additional 50 cases in the medical literature.

In our cohort, all were women, and the age of onset was at 10, 46 and 68 years. The initial symptom was abdominal pain, and all had two or more episodes of DH that required at least one surgery, as well as intensive treatment with corticosteroids, immunosuppressants or IVIG. The underlying lesion was a perforated ulcus, and pathology showed typical images of vascular ectasia. Close observation and repeated studies were often necessary to reach a diagnosis. All died, two from complications related to the GI involvement and one of sudden death (Appendix A).

In the published cases, more than three fourths were women. A clear age predominance could not be described, but there is a tendency to occur in the elderly and in children. The onset of clinical manifestations was acute or subacute in all patients, and the duration of symptoms from baseline to diagnosis confirmation ranged from 2 days to 6 months. Juvenile DM patients seemed to show GI involvement earlier than adult during their disease course. The usual clinical presentation included abdominal pain (65%), diarrhoea (14.5%), vomiting (16.6%), fever and HD (12.5%) (Figure 3). All patients presented muscular and cutaneous involvement at the time of diagnosis. Although muscle biomarkers were barely described, raised serum CK levels were detected in 7 out of 10. Endoscopic studies were performed in the 31.2% of the patients, and the underlying lesion was spontaneous perforation or ulcer in 45%. The upper and lower GI tract were equally affected without a clear trend, but when perforation occurred duodenum was the most common site. Moreover, duodenum perforation seemed to occur more often in juvenile patients, and on the other hand, PI seemed to occur more often in adult DM patients. This fact has not been described previously, and it is unknown if there is an explanation for it or if it is mere coincidence given that it is a small number of cases.

Furthermore, it would have been of interest to study the possible association between GI involvement and interstitial lung disease or malignancy, but such information was hardly found in case reports. The antibody panel was not described in most of the cases, especially in older ones, but many of them were positive for NXP-2, as has been described in other reports [8]. This aspect is of special interest as myositis-specific autoantibodies (MSA) are increasingly used to delineate distinct subgroups of DM, but to date, the role of MSA in DM presenting with GI complications has not been reported. In relation to histopathology, vasculitis was found in 13 cases. With respect to treatment, high doses of corticosteroids (1 mg/kg/day) were the rule (*n* = 19), but 20.8% required IGIV at some point, and 33% underwent surgical treatment. Mortality was 41.7%, and the most frequent cause of death was massive bleeding.

Early identification of GI involvement may be challenging, since symptoms often are nonspecific and progressive, and in some cases long lasting. Special attention should be paid to disease activity in its classical manifestations, especially skin and muscle activity, since it has been reported to be present in almost all patients at the time of GI involvement.

In untreated patients, an acute inflammatory arteriopathy with intimal hyperplasia of arteries and veins and vessel occlusion by fibrinous thrombi in the submucosa, muscle and adjacent serosa has been described [4]. In contrast, patients under treatment showed a chronic form of noninflammatory arteriopathy, characterised by narrowing of multiple small and medium arteries, subintimal foam cells, fibromyxoid neointimal expansion and significant luminal compromise with macrophage infiltration through the muscle into the intima. These changes may occasionally also be observed in both muscle and skin. This chronic arteriopathy appears to be the underlying pathogenesis of the GI ulceration or perforation, but also may be the consequence of the DM therapy [9]. On the other hand, well documented cases of true vasculitis in GI complications are extremely rare and occur more commonly in juvenile DM [10,11,12,13,14].

The key role of vascular compromise in the physiopathology of GI tract involvement has been suggested in several studies [10]. The activation of complement and the formation of membrane attack complex (MAC), which is deposited on the endomysial capillaries, leads to lysis of endothelial cells and capillary destruction. Loss of capillaries results in smooth muscle ischaemia [12], as shown in muscle biopsy from our case one (Figure 1A). Additionally, release of von Willebrand factor due to MAC deposits promotes an inflammatory process, leading to thrombus formation and vascular obliteration, and causes ischemia and necrosis [6,13,14]. Von Willebrand factor VIII-related antigen has been described as a laboratory test sometimes used as a measure of juvenile DM disease activity, suggesting endothelial activation [13].

With respect to histopathological features in our cases, we observed ischemic vascular disease with mucosal inflammation, granulation tissue replacement and submucosal oedema in areas, with significant destruction of the muscle layer. These findings suggest that the underlying pathophysiological mechanism could be a vasculitis or other microvascular disease, as occurs in the typical muscle DM involvement. It is probable that the same mechanism which affects muscle, also affects the skin, lungs, and gastrointestinal tract. It is currently known that certain specific antibodies mainly affect the skin and/or lung. But in the case of gastrointestinal involvement, it could be related to the anti-NXP2 antibody as mentioned previously, or it could only be subject to a severe effect of the disease. In our experience, in recent years, this complication has not been observed, so it is more likely that it has to do with poor disease control than with an association with a specific antibody.

Regarding the clinical scenario of IP, this condition has been rarely described in adult patients with DM, and several mechanisms have been proposed for its development, including breaks in the intestinal mucosa, infection and ischaemia due to vasculitis. It is often an asymptomatic condition and may occur with pneumoperitoneum, which is sterile and must be distinguished from a perforated viscus [12].

Given the heterogeneous presentation of DM, treatment often has to be specific to organ manifestations or associated conditions. Corticosteroids constitute the first choice for initial treatment. One important consideration is the fact that GI absorption of corticosteroids or other drugs can be variable when patients have associated GI vasculopathy, so high-dose intravenous methylprednisolone pulses may be considered at the beginning. However, longstanding treatment with high doses of corticosteroids could also be a predisposing factor for the development of GI complications [9]. Nonetheless, since GI compromise may be life-threatening, an aggressive treatment, including additional immunosuppressive drugs and/or IGIV, should be considered. Moreover, surgery can be necessary in refractory cases. Several studies including DM patients with GI tract involvement suggest that methotrexate and azathioprine may be a good treatment option [5,13,14].

Overall, this study has a main limitation: a risk of bias associated with case reports due to the heterogeneity of the cases presented. In addition, due to the retrospective nature of this work and as some of the patients were treated years ago, some aspects, such as immunology, that would have been of interest remain unknown.

## 5. Conclusions

GI involvement other than dysphagia are relatively rare in DM, but they can be a life-threatening condition. Female, child-youth age and acute or subacute abdominal pain along with muscular and cutaneous involvement are some clinical-epidemiological features that would allow an early diagnosis and treatment to improve the prognosis. We still don’t know if the underlying pathophysiological mechanism is a true vasculitis or other kind of vascular disease as seen in the muscle tissue. Since knowledge of the types of vasculopathy that may involve the GI tract is key to achieving a correct diagnosis, studies with a larger number of patients are needed. Further studies are required to assess new tools for treatment which can modify the disease course and improve the clinical outcomes.

## Figures and Tables

**Figure 1 diagnostics-12-01200-f001:**
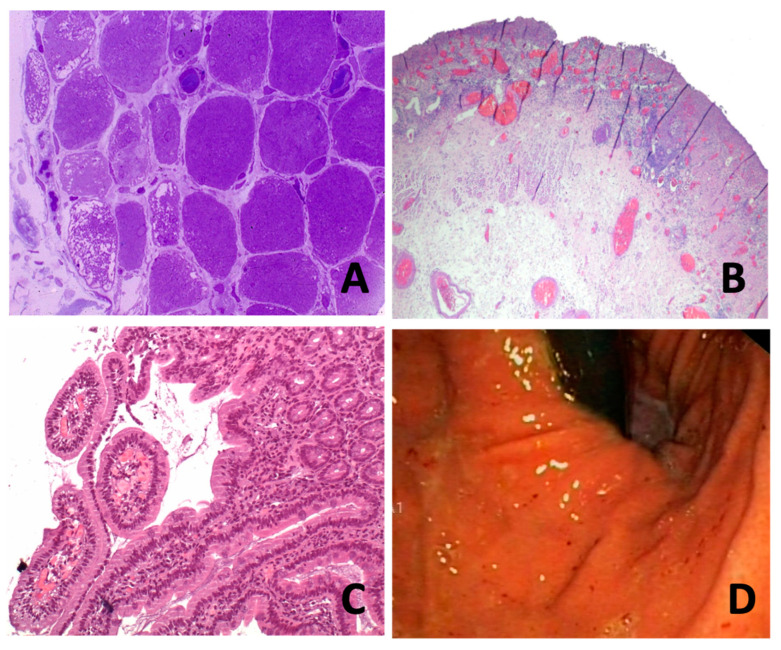
(**A**) (case 1): Plastic-embedded semithin cross-section muscle biopsy. Note the loss of capillaries in the peripheral area with mega-*capillaries (star).* (**B**) (case 2): Ulcerated mucosa area lined by granulation tissue, vascular ectasia and acute fibrinoleukocyitary inflammation. Some areas of the oesophageal wall with a moderate atrophy of the inner muscular layer with slight fibrosis and oedema. (**C**) (case 2) Mucosal inflammatory infiltrate with vascular ectasia extending to submucosa. Decreased glandular acini occupied by fibrin. (**D**) (case 3): Antral staking erythematosus without erosions or ulcerations evident. In the duodenum (from bulb to third duodenal portion), fibrin-coated erosions and well demarcated erythematous lesion with peripheral oedema is seen, without evidence of active bleeding.

**Figure 2 diagnostics-12-01200-f002:**
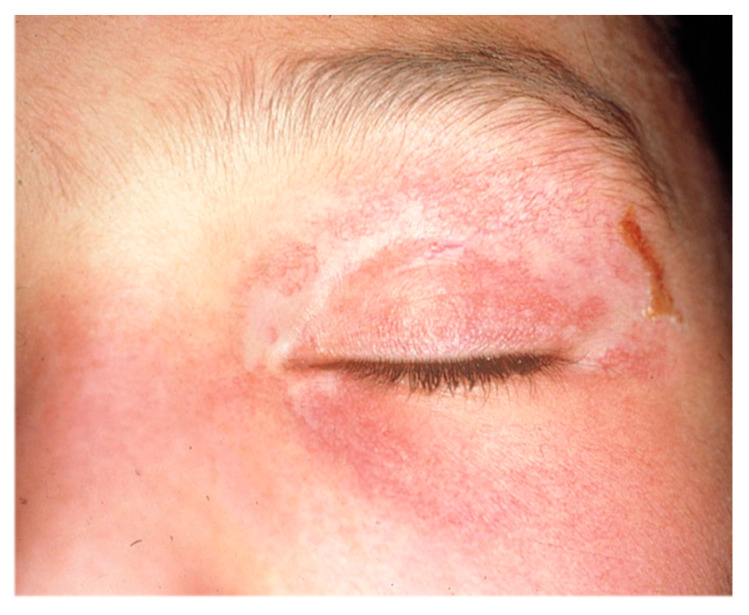
Appearance of the heliotrope rash coinciding with the onset of digestive symptoms in *case 1*.

**Figure 3 diagnostics-12-01200-f003:**
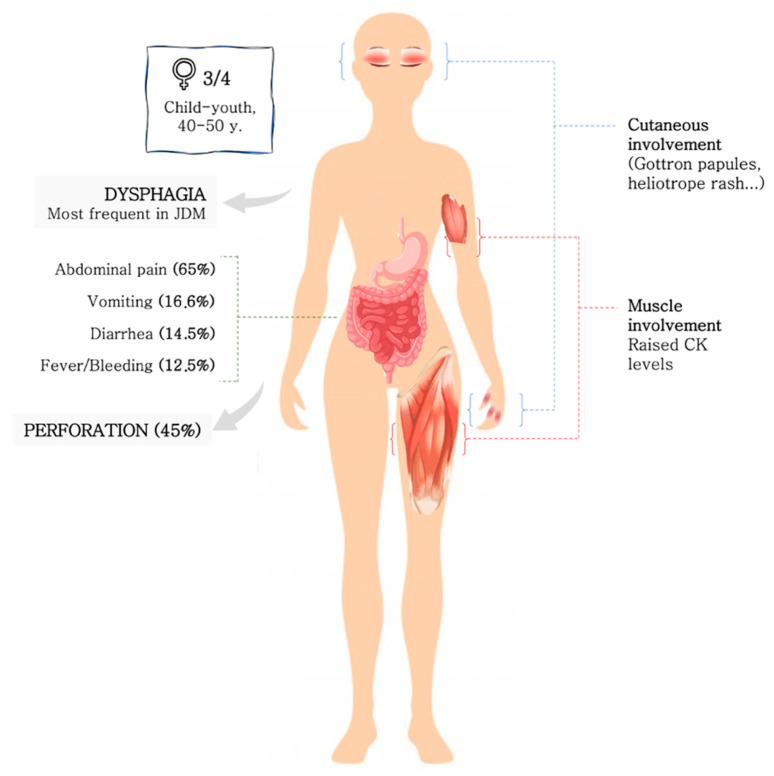
Clinical-epidemiological features of GI involvement in DM.

**Table 1 diagnostics-12-01200-t001:** Characteristics of patients with DM and severe GI compromise from our unit at the Hospital Clínic de Barcelona.

	Age-Gender	DM Muscle Biopsy	Treatment	Outcome	Time to Perforation	GI Symptoms	Muscular/Skin Activity	Affected Areas	Endoscopic Appearance	Pathological Anatomy	Response to Treatment
1	10-F	+	PDN, CYC, cloroquine, IVIG	Death	1 week	Abdominal pain, dysphagia, GI bleeding	+/+	Oesophagus, stomach, duodenum	Erosions, oesophageal ulcers, inflamed mucosa	Vascular changes consistent with vasculitis	Partial
2	68-F	+	PDN + IVIG	Death	4 week	Diarrhoea, abdominal pain, GI bleeding	+/-	Oesophagus, stomach, ileocecal valve	Erosions, oesophageal ulcers, inflamed mucosa with active bleeding	Oesophagus with moderate wall inner muscular atrophy with fibrosis and oedema	-
3	46-F	+	PDN + AZA + IVIG	Death	3 week	Abdominal pain, GI bleeding	+/+	Stomach, duodenum	Erosions, oesophageal inflamed mucosa, ulcers and petechiae	Fibrosis without atrophy and marked ectasia	+

## Data Availability

The authors confirm that the data supporting the findings of this study are available within the article and its Appendix A.

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
