# Peer review of "Gastrointestinal Involvement in Dermatomyositis"

_diagnostics, 2022, doi:10.3390/diagnostics12051200_

Round 1

Reviewer 1 Report

The authors report three cases of GI involvements other than dysphagia that they have experienced at the Hospital Clínic de Barcelona. This is extremely rare complication and difficult to notice even for experienced rheumatologists or dermatologists. On the other hand, it also shows that GI is a very deadly complication. Together with previous reports, the clinical background of DM with GI, consideration of the mechanism of GI onset, and the importance of stronger and earlier therapeutic interventions are shown. These information are very useful for clinicians.

Author Response

Response to reviewer.

On behalf of all the authors of the article, we would like to thank the reviewers for their corrections and contributions that improve the edition and its quality.

Reviewer 1:

  1. Extensive editing of English language and style required.

An exhaustive edition of the English and format has been made. A description of the corrections made has been included as part of the change control.

Reviewer 2 Report

Matas-Garcia et al. present three cases of fatal gastrointestinal tract (GIT) involvement in dermatomyositis (DM) and discuss their experience in the context of fifty more cases published in the literature.

Following comments:

Pinpointing gastrointestinal involvement in DM is important, yet the relevance not really supported by this publication. The authors state to only refer to fatal disease which seems very rare. Any GIT involvement would have been more interesting. Are post mortem data available similar to systemic scleroderma where GIT involvement of any severity may be seen in more than 70% of cases.

The authors describe a considerable number of DM patients in their center, yet only 3 had fatal involvement of GIT, one of them most probably not related to DM. Is this really relevant?

Have in the three presented cases DM-specific antibodies been detected?

Case 2: what is the stage of tumor disease? Was there metastatic disease?

Which and how many of cases in Supplemental Table S1 are paraneoplastic?  

Table 1: time to perforation is given in only weeks, yet in the text intervals seem much longer.

Do the authors wish to used IVIG or IGIV for intravenous immunoglobulins? Use one acronym consistently.

Page 6, line 203: what does HD mean?

Page 7, line 210: what does PI mean?

Figure S1: are the publications studies or case reports?

As indicated by authors, prospective studies indeed seem mandatory to settle the importance of GIT involvement.

Author Response

Response to reviewers.

On behalf of all the authors of the article, we would like to thank the reviewers for their corrections and contributions that improve the edition and its quality.

Reviewer 2:

  1. English language and style are fine/minor spell check require

An exhaustive edition of the English and format has been made.

  1. Pinpointing gastrointestinal involvement in DM is important, yet the relevance not really supported by this publication. The authors state to only refer to fatal disease which seems very rare. Any GIT involvement would have been more interesting. Are post mortem data available similar to systemic scleroderma where GIT involvement of any severity may be seen in more than 70% of cases.

Thank you very much for the comment. Unfortunately, we do not have post-mortem data. As you say and we reflect in this article, gastrointestinal involvement is rare. We understand that this happens due to a more severe course of the disease and systemic involvement. Currently, refractory cases are rare, and patients start appropriate treatment earlier, which would considerably reduce the development of gastrointestinal tract involvement.

  1. The authors describe a considerable number of DM patients in their center, yet only 3 had fatal involvement of GIT, one of them most probably not related to DM. Is this really relevant? Have in the three presented cases DM-specific antibodies been detected?

Thank you very much for your comment. As you rightly say, these are rare cases in the context of a rare disease. Even worse, I think it is important to report it since it allows us to better understand this disease, which does not only affect muscle and skin. We understand that according to the severity of it other organs are affected, as in this case the gastrointestinal tract. Although the pathophysiological mechanism is unknown, the initially damaged structures are likely the capillaries.

These cases are old, and at that time we could not assess the specific antibodies. We have added to the text, in the discussion part, this aspect as a limitation

  1. Case 2: what is the stage of tumor disease? Was there metastatic disease?

Colon cancer was localized, and the patient had no recurrence after specific treatment. This information has been added to the article.

  1. Which and how many of cases in Supplemental Table S1 are paraneoplastic?

This information was not systematically analyzed since it did not appear in many of the reported cases.

  1. Table 1: time to perforation is given in only weeks, yet in the text intervals seem much longer.

Thank you very much for the appreciation. Indeed, the times in the table were misleading. The time to perforation referred to the time elapsed from hospital admission at the first symptom to the detection of said complication. The times reflected in the table are correct. However, it has been clearly worded in the manuscript to avoid confusion.

  1. Do the authors wish to used IVIG or IGIV for intravenous immunoglobulins? Use one acronym consistently.

Thank you very much for the comment. We will use IVIG for intravenous immunoglobulins.

  1. Page 6, line 203: what does HD mean?

It is digestive haemorrhage (DH). It has been changed in the text.

  1. Page 7, line 210: what does PI mean?

It is intestinal pneumatosis (IP). It has been modified in the text.

  1. Figure S1: are the publications studies or case reports?

All cases included in the supplementary table are case reports or case series. The largest of the case series included 4 patients and was published in 1988. This information has been added to the article, in the material and methods part.
